# Outcome-based Semifactual Explanation For Reinforcement Learning

## Abstract

Counterfactual explanations in reinforcement learning (RL) aim to answer what-if questions by showing sparse and minimal changes to states, which results in the probability mass moving from one action to another. Although these explanations are effective in classification tasks that look for the presence of concepts, RL brings new challenges that current counterfactual methods for RL still need to solve. These challenges include defining similarity in RL, out-of-distribution states, and lack of discriminative power. Given a state of interest called the query state, we solve these problems by asking how long the agent can execute the query state action without incurring a negative outcome regarding the expected return. We coin this outcome-based semifactual (OSF) explanation and find the OSF state by simulating trajectories from the query state. The last state in a subtrajectory where we can take the same action as in the query state without incurring a negative outcome is the OSF state. This state is discriminative, plausible, and similar to the query state. It abstracts away unimportant action switching with little explanatory value and shows the boundary between positive and negative outcomes. Qualitatively, we show that our method explains when it is necessary to switch actions. As a result, it is easier to understand the agent's behavior. Quantitatively, we demonstrate that our method can increase policy performance and, at the same time, reduce how often the agent switches its action across six environments. The code and trained models are available at https://anonymous.4open.science/r/osf-explanation-for-rl-E312/.

## 1 Introduction

Reinforcement learning (RL) has shown incredible performance in several domains. These include surpassing human performance in various games like Go, chess, poker, Atari (Mnih et al., 2013; Schrittwieser et al., 2020; Silver et al., 2016; Brown & Sandholm, 2017), robotics (Tang et al., 2024), and healthcare (Yu et al., 2023). Much of these accomplishments are due to neural networks being flexible function approximators. However, the flexibility is obtained by sacrificing other desirable properties, such as explainability (Arrieta et al., 2020). Explainability is crucial since it helps stakeholders, from the end users to researchers, understand and trust RL systems. By understanding the systems, they can be improved, corrected, and be safely deployed. Furthermore, they can be used in high-stake domains such as healthcare, criminal justice, and finance (Yang et al., 2023).

Explainability in RL has become increasingly popular recently and resulted in the research field known as explainable reinforcement learning (XRL) (Qing et al., 2023; Glanois et al., 2024; Amitai & Amir, 2024; Milani et al., 2024; Hickling et al., 2024; Bekkemoen, 2024). The XRL community has mainly focused on leveraging methods from the supervised learning explainability field and often does not consider the specific challenges that make explainability in RL difficult. These challenges include delayed rewards, stochastic environments, and large state spaces (Amitai & Amir, 2024). For example, feature importance is one of the most popular methods in XRL but does not specifically tackle sequential decision-making (Bekkemoen, 2024). We need additional explainability tools tailored towards RL to better understand the behavior in sequential decision-making.

Counterfactual explanations are relatively new in XRL and can potentially help us understand RL agents. However, the existing counterfactual methods in RL have problems, which we illustrate with an example from Huber et al. (2023). In Fig. 1, we notice two problems with a generative approach to

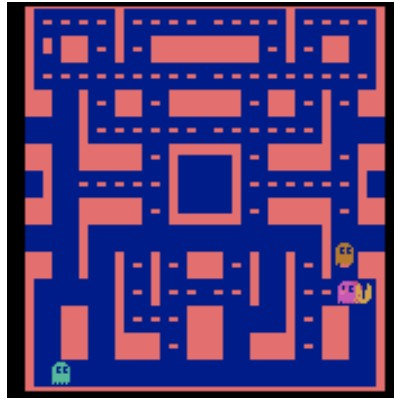 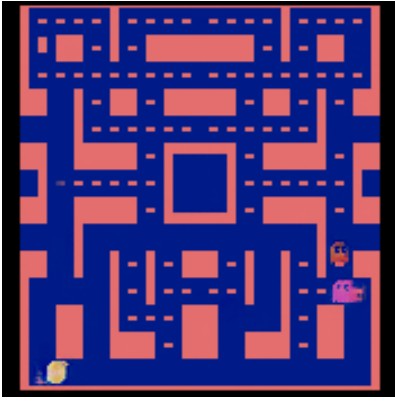

(a) **Query state.** The agent moves down.     (b) **Counterfactual state.** The agent moves up.

Figure 1: Example of a counterfactual explanation in Ms. Pac-Man. The game screens are from the arXiv version of Huber et al. (2023, Figure 3), licensed under CC BY 4.0 (`https://creativecommons.org/licenses/by/4.0/`).

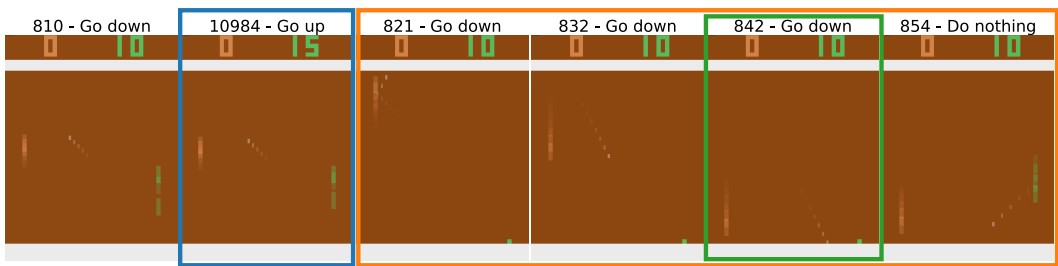

Figure 2: The query state is shown in the timestep 810, counterfactual state (blue) in timestep 10984, and outcome-based semifactual (OSF) state (green) in timestep 842. The trajectory from the query state to the OSF state and the state after is shown in the orange box. The top row shows both the timestep and the action taken in the state. The counterfactual state is the closest state found via the nearest neighbor search where the distance is measured in the policy's embedding space. The nearest neighbor is selected from a replay buffer of 100 000 states. The timestep counter is not reset when an episode ends, thus, the timestep count for the counterfactual state is large.

creating counterfactual states. First, the generated game objects like Pacman, ghosts, and some pills look unrealistic, creating out-of-distribution states. Consequently, the generated states can decrease the trust as they do not convince or satisfy the stakeholders' expectations. Second, the counterfactual states should be close to the query states. However, in contrast to supervised learning, the notion of similarity in RL is more challenging to define. Therefore, states that are visually similar do not necessarily imply that the states are close. The states might not be reachable, they can occur far from each other in time, and small changes can significantly impact the game semantics. In Fig. 1, the agent itself is moved, making it difficult to understand what game elements made the agent choose to go down. An retrieval-based approach using a replay buffer can solve the first problem with out-of-distribution states. However, the states we retrieve using a replay buffer can be too visually similar and do not have the discriminative power to convey the agent's behavior as seen in Fig. 2.

Semifactual explanations describe how much we can modify input features without changing the agent's action (Aryal & Keane, 2024). Semifactual states are states furthest away from the query state without crossing the decision boundary, shown in Fig. 3a, but suffer from a lack of discriminative power like counterfactual explanations. To alleviate the problems mentioned, we propose the outcome-based semifactual (OSF) explanation and introduce a simulation-based approach to find OSF states to explain the agent's behavior. Our OSF explanations differ from semifactual explanations used in supervised learning, we aim to look at it from the value space perspective rather than looking at the probability distribution over actions. Given a query state and simulated trajectories from it, we ask how long we can keep executing the query state action and expect a similar (but

not necessarily optimal) outcome. For example, Fig. 3b illustrates a OSF state in the mountain car environment where we keep accelerating left beyond the decision boundary while still being able to reach the goal without the agent taking an additional lap. Fig. 2 illustrates an OSF state and a counterfactual state with respect to a query state in Pong. The OSF state show us the boundary between positive and negative outcomes rather than a single step decision boundary. Additionally, the OSF state reduces the amount of action switching the agent makes by abstracting away unimportant behavior.

In summary, we propose: 1) Outcome-based semifactual (OSF) explanations that focus on the outcome of an agent. They enable us to understand which action switches are critical to the agent and which can be abstracted away. Furthermore, they are important states on the boundary between positive and negative outcomes that further help us understand the agent. 2) We show the effectiveness of OSF explanations via various case studies and compare them to counterfactual explanations using the nearest neighbor search. 3) We demonstrate that we can use OSF explanations to minimize the number of times the agent switches action during rollouts. Furthermore, we show that this minimization can improve policy performance in six environments while abstracting away unimportant behavior with little explanatory value.

## 2 BACKGROUND

**Reinforcement Learning.** In RL, we model the environment as a Markov decision process (MDP) consisting of a tuple $\langle \mathcal{S}, \mathcal{A}, T, R, \gamma \rangle$ ( Sutton & Barto, 2018; Achiam, 2018; Chapter 2 in Albrecht et al., 2024). $\mathcal{S}$ is the set of states, $\mathcal{A}$ is the set of actions, $T : \mathcal{S} \times \mathcal{A} \times \mathcal{S} \to [0,1]$ is the state transition probability function, $R : \mathcal{S} \times \mathcal{A} \times \mathcal{S} \to \mathbb{R}$ is the reward function, and $\gamma \in [0,1]$ is the discount factor. Additionally, we have the starting state distribution defined by $\mu : \mathcal{S} \to [0,1]$. An agent interacts with the environment via a policy $\pi : \mathcal{S} \to \mathcal{A}$ that takes a state and outputs a corresponding action. The policy can either output a distribution over actions or be indirectly defined via a state-action value function. The trajectory is defined by $\tau = (s_1, a_1, s_2, a_2, \ldots)$ where $s_1 \sim \mu$. The probability distribution over trajectories conditioned on the policy $\pi$ is defined by $Pr(\tau|\pi) = \mu(s_1) \prod_{t=1} T(s_{t+1}|s_t, a_t)\pi(a_t|s_t)$. We use value functions to measure the expected returns, which are state-based or state-action-based. The expected return starting from the state $s_1$ and following the policy thereafter is defined by $\mathbb{E}_{\tau \sim Pr(\cdot|\pi)}\big[\sum_{t=1}^{\infty} \gamma^{t-1} R(s_t, a_t, s_{t+1})\big]$. The state value function is defined via the Bellman equation by $V^\pi(s) = \mathbb{E}_{s' \sim T(\cdot|s,a), a \sim \pi(\cdot|s)}[R(s, a, s') + \gamma V^\pi(s')]$. Similarly, the state-action value function known as the Q-function is defined via the Bellman equation by $Q^\pi(s, a) = \mathbb{E}_{s' \sim T(\cdot|s,a)}\big[R(s, a, s') + \gamma \mathbb{E}_{a' \sim \pi(\cdot|s')}[Q^\pi(s', a')]\big]$. Our work requires access to a policy modeled via the Q-function to measure the expected return. Specifically, we use the deep Q-network (DQN) (Mnih et al., 2013) but any Q-learning methods work.

**Counterfactual and Semifactual Explanation.** We define a counterfactual explanation with respect to a policy $\pi$ as a tuple $\langle s, s' \rangle$ where $s$ is the query state and $s'$ is the counterfactual state (Guidotti, 2022). Given the policy $\pi$ and query state $s$, we get $\pi(s) = a$. The counterfactual state $s'$ is similar to $s$ but where the features have small sparse changes such that $\pi(s') = a' \neq a$. The "optimal" counterfactual state has certain desirable properties such as validity, sparse changes with respect to the query state, and similarity. A semifactual explanation is similar where the features are altered but where the query state $s$ and the semifactual state $s'$ maintain the same action, $\pi(s') = a = \pi(s)$. We want the change between the query and semifactual states to be as large as possible to observe the states' differences while not crossing the decision boundary. Like counterfactual explanations, semifactual explanations should satisfy certain desirable properties (Aryal & Keane, 2023). Figure 3a shows an example of a query state, a counterfactual state, and a semifactual state.

## 3 RELATED WORK

There are several taxonomies for XRL, each with its strengths and weaknesses (Qing et al., 2023; Glanois et al., 2024; Amitai & Amir, 2024; Milani et al., 2024; Hickling et al., 2024; Bekkemoen, 2024). Our work is motivated and inspired by methods that fall within the post hoc explainability category. Specifically, our method is motivated and inspired by counterfactual explanation methods

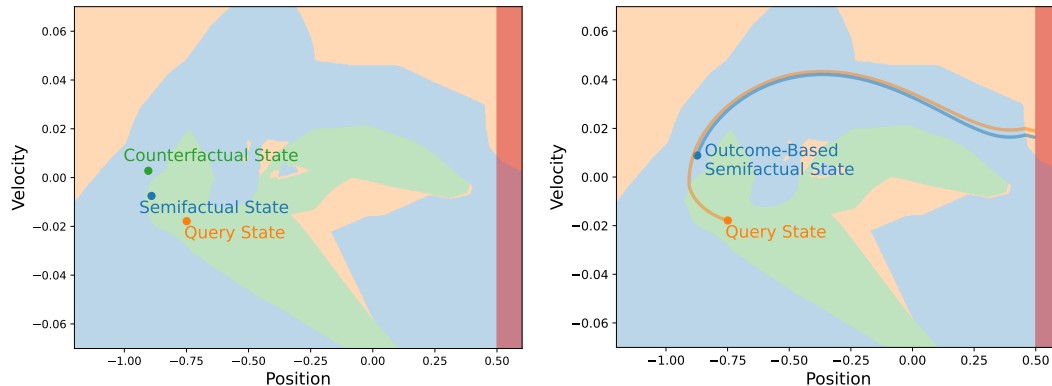

(a) Example of the states: query, counterfactual and semifactual. These states are for illustrative purposes only. They are manually selected from a trajectory and not found via an algorithm. The figure is inspired by Aryal & Keane (2024, Figure 1).

(b) Example of the states: query and OSF. The OSF state is produced by simulating from the query state and is found via our algorithm. The lines show trajectories created by starting from the two states and following the policy thereafter.

Figure 3: State examples in the mountain car environment (Moore, 1990; Towers et al., 2024). A pretrained Q-network by RL Baselines3 Zoo (Raffin, 2020) is used as the policy. The red area highlights the goal of the environment. Green area = accelerate left , orange area = do nothing , and blue area = accelerate right .

for RL and state importance methods that explain by finding and presenting important or critical states to stakeholders. Below, we look at each of these areas of work in more detail.

**State Importance.** State importance methods are among the most popular in XRL. As far as we know, Amir & Amir (2018) and Huang et al. (2018) proposed the earliest work in this category of methods. They show stakeholders a summary of important states to understand an agent's global behavior. The importance definition varies between studies. Some studies use Q-values to measure state importance, while others use quantities like interestingness (Sequeira & Gervasio, 2020). User studies have shown that these explanations are effective at aligning mental models (Huber et al., 2021; Amitai & Amir, 2024). One of their strengths is their ease of use and applicability. They discover and find these important states via simulations. The downside is the need for a state transition model, which can partially be fixed by learning world models (Ha & Schmidhuber, 2018). New importance measures have been proposed in later studies (Huang et al., 2019; Lage et al., 2019; Sequeira & Gervasio, 2020). Our method similarly leverages Q-values as an importance measure. It uses a state transition model like the methods above. However, our goal is to abstract away complex agent behavior, and find states that are important and on the boundary between positive and negative outcomes.

**Counterfactual Explanations in RL.** Both Olson et al. (2021) and Huber et al. (2023) use generative adversarial networks to generate counterfactual states. In the introduction, we mentioned problems with these approaches, namely out-of-distribution states, how to handle the similarity measure, and a lack of discriminative power. To solve these problems, we improve upon these counterfactual methods by proposing a different type of explanation, namely OSF explanations.

**Semifactual Explanations.** Semifactual explanations are not new and have been researched in the context of supervised learning (Kenny & Keane, 2021; Kenny & Huang, 2023). However, there has been little work on semifactual explanations in RL (Gajcin et al., 2024). Semifactual explanations deal with even-if explanations (Aryal & Keane, 2024). For instance, imagine we have a system that predicts the risk of diabetes. An explanation could be that a person has a low risk of diabetes according to the system and that even if they doubled the sugar intake, the risk would still be low. This is a semifactual explanation since the decision stays the same while the input changes. We extend semifactual explanations and use OSF explanations to explain the outcome of actions as seen in Fig. 3b.

## 4 METHOD

We introduce the concept of outcome-based semifactual (OSF) explanations and how we can find OSF states algorithmically.

**What is an OSF explanation?** An OSF explanation consists of a tuple $\langle s_t, s'_{t+n}, \delta \rangle$ that is constructed with respect to a policy. $s_t$ is the query state, $s'_{t+n}$ is the OSF state, and $\delta$ is the stopping criterion. We assume the policy is represented as a Q-function. Given $a_t = \arg\max_a Q(s_t, a)$, $s'_{t+n}$ is the state $n$ timesteps after $s_t$ by only executing the action $a_t$ and where $\max_a Q(s_{t+n}, a) - Q(s'_{t+n}, a_t) \leq \delta$. $s_{t+n}$ is the state $n$ timesteps after $s_t$ by following the policy. Because $s'_{t+n}$ is produced by a rollout from $s_t$, we assume that changes in features are sparse. Additionally, the OSF state is visually similar to the query state given that $\delta$ is not too large.

**Finding an OSF state.** We want to understand how much a query state $s_t$ can be modified by executing the same action without affecting the future outcome negatively. We define negative outcomes as states where the return is significantly lower than what is achieved by following the policy. We seek an OSF state, as seen in Fig. 3b. To find the OSF state, we assume access to a Q-function and a state transition probability function $T$ that enables us to simulate trajectories. OSF states we find depend on whether the state transition probability function is stochastic or deterministic. The experiments use deterministic state transition probability functions and are therefore unaffected by stochasticity. Using the Q-function, we define the importance gap between two states $s$ and $s'$ conditioned on the action $a$ by

$$IG(s, s' \mid a) = \max_{a'} Q(s, a') - Q(s', a). \tag{1}$$

The method starts by first following the policy from the state $s_t$ to find the trajectory $\tau = (s_t, a_t, s_{t+1}, a_{t+1}, \ldots, s_N)$. After obtaining the trajectory $\tau$, we simulate a new trajectory from the state $s_t$, but instead of following the policy, we execute the action $a_t = \arg\max_a Q(s_t, a)$ to all states that follow and obtain the trajectory $\tau' = (s_t, a_t, s'_{t+1}, a_t, s'_{t+2}, a_t, \ldots, s'_T)$. The OSF state is found by applying the following criterion

$$n = (\arg\min_i IG(s_{t+i}, s'_{t+i} \mid a_t) > \delta) - 1 \tag{2}$$

to the trajectories. Hence, $s'_{t+n}$ is the OSF state. The criterion says to keep executing action $a_t$ as long as the gap is lesser than or equal to $\delta$. The criterion only needs to be one-sided since a negative gap tells us we are reaching states that are better than following the policy. The gap will generally be positive as the policy is "optimal". We allow a small gap since a small loss in expected return will not affect the outcome negatively. For example, a small $\delta$ will still allow the policy in mountain car to reach the goal, albeit taking a few more timesteps.

The $\delta$ value depends on the environment because the expected return varies based on the reward, which differs between environments. On the one hand, a small $\delta$ value results in the query state and OSF state being very similar and lowers the discriminative power. On the other hand, a large $\delta$ value makes the OSF state dissimilar to the query state and ends up being a state that the policy rarely encounters. The $\delta$ value has to be set by a human stakeholder that interactively explores different $\delta$ values.

## 5 EXPERIMENTS

We detail the experimental setup needed to reproduce our results and give qualitative and quantitative evidence of the method's effectiveness in mountain car and several Atari environments.

### 5.1 EXPERIMENTAL SETUP

The mountain car policy is a pre-trained DQN by Raffin (2020). The DQN policies used in Atari environments are trained using CleanRL (Bellemare et al., 2013; Huang et al., 2022). We use the default hyperparameters used by CleanRL in commit `65789babaae033433078504b4ff0b925d5e27b99` for all Atari environments. All the environments are implemented in Gymnasium v0.29.1 (Towers et al., 2024). Our method only has one hyperparameter, which is $\delta$ that we document separately for each experiment. We overlay observations from 9 timesteps before the indicated timestep to show movements leading up to the state

shown in the figures. The quantitative results shown in Table 1 are computed by averaging across 30 episodes, where the environments are initialized with a new seed for each episode.

We provide the code and the trained models at `https://anonymous.4open.science/r/osf-explanation-for-rl-E312/`. The data used is generated on the fly with Gymnasium. All experiments ran on a MacBook Pro 2023, Apple M2 MAX, 64 GB RAM. We used Python v3.11.8 and PyTorch v2.2.2 (Ansel et al., 2024) with the Metal Performance Shaders (MPS) backend for graphics processing unit (GPU) accelerated training. The package installer for Python (PIP) requirement file in the link above provides the complete list of packages and their versions.

## 5.2 CASE STUDIES

We look at OSF explanations for two query states across three environments: mountain car, Pong, and Breakout. For Pong and Breakout, in addition to OSF explanations, we inspect the counterfactual explanations for the same query states. The counterfactual states are found by looking for the closest states based on the policy's embedding space. To accomplish that, we generate 100 000 states from simulations to find the closest states. From those 100 000, we select the 100 closest states and sample 4 states from those 100 at random to avoid selecting states too visually similar. These 4 are visualized as the counterfactual states in Figs. 6, 8, 10 and 12.

### 5.2.1 MOUNTAIN CAR

In the first case study, we look at specific parts of the decision boundary that are unsmooth in the mountain car environment. We ask why they exist and whether they are necessary. Since the mountain car environment only has two features, we can easily visualize its state space and the corresponding decision boundaries.

In Fig. 4a, the query state is in a region where the policy accelerates right but changes quickly to do nothing before changing to accelerate left. In this specific case, we see that the OSF state is in the accelerate left region of the state space. This means we can skip the do nothing action and keep accelerating right until gravity pulls the car into the accelerate left region. By only using the query state action, the agent can reach the goal quicker than by following the policy. It is difficult to conclude that the do nothing region is useless. However, in this case, the complexity of the decision boundary can be reduced.

Fig. 4a shows that specific parts of the orange region are not necessary, while Figs. 4b and 4d depict that other parts are needed. Figs. 4b and 4d show that the orange region helps reduce the car's acceleration when going towards left. Although we want the car to get high up on the left to get enough speed to go right, too high is unnecessary. Going too high up to the left wastes time because the car can get to the goal with less speed. Fig. 4c shows the small blue island next to the query state is unneeded as the OSF trajectory after the island is similar to the original trajectory. Figs. 4e and 4f show two situations where a small patch of orange area next to the query state decreases the time it takes to reach the goal if we use it and increases the time if we do not use it.

To conclude, we have observed that the complexity of the decision boundaries are sometimes needed. While in other cases, they add unnecessary complexity without increasing the policy's performance.

### 5.2.2 PONG

For Pong, we look at a query state from an episode at timestep 200. Fig. 5 shows the query state at timestep 200 and the corresponding OSF explanation. The first row in the figure displays how the policy behaves from timestep 200 to 220. We observe the policy is making several moves, which can be unintuitive for humans as the ball is far away, and thus, those moves are unecessary and convey little explanatory value. On the second row, we observe that the OSF explanation tells us that none of the moves the policy is making are necessary. The policy only needs to move into a receiving position after timestep 213, which is the OSF state. The OSF explanation simplifies the policy's movements so that we can focus on important moves the policy needs to make. Moreover, the OSF state is the last chance to move into a receiving position, indicating the boundary between outcomes.

Fig. 6 shows counterfactual explanations for the same query state. The counterfactual explanations indicate that the policy focuses on the ball and the opponent's paddle when representing a state

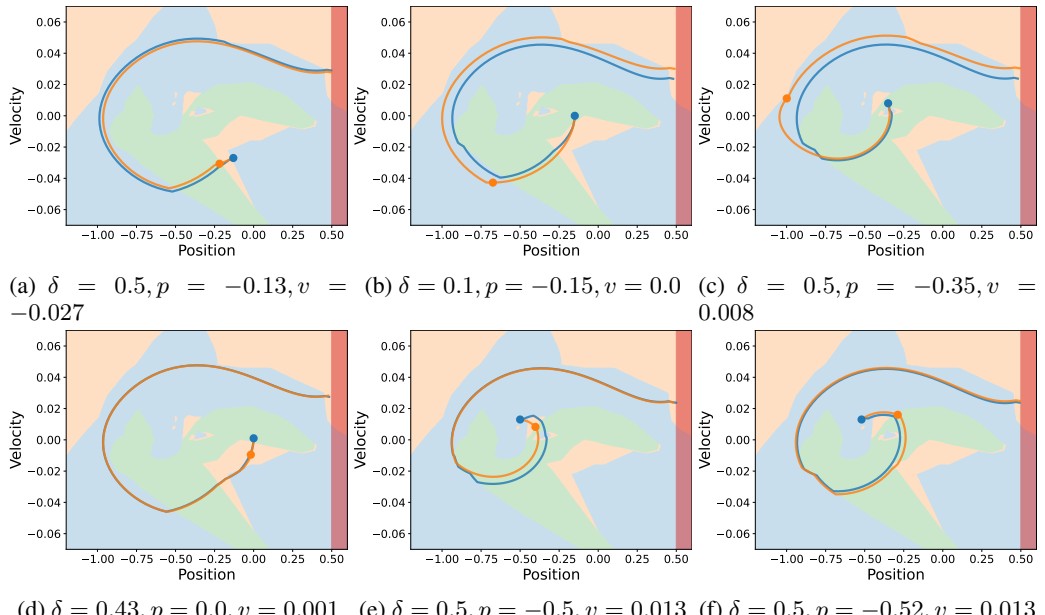

(a) $\delta = 0.5, p = -0.13, v = -0.027$

(b) $\delta = 0.1, p = -0.15, v = 0.0$

(c) $\delta = 0.5, p = -0.35, v = 0.008$

(d) $\delta = 0.43, p = 0.0, v = 0.001$

(e) $\delta = 0.5, p = -0.5, v = 0.013$

(f) $\delta = 0.5, p = -0.52, v = 0.013$

Figure 4: **Mountain Car.** Query states (blue markers) and corresponding OSF states (orange markers). $\delta$ is the stopping criterion, $p$ indicates position, and $v$ is the velocity of the query state. The red area highlights the goal of the environment. Green area = accelerate left, orange area = do nothing, and blue area = accelerate right.

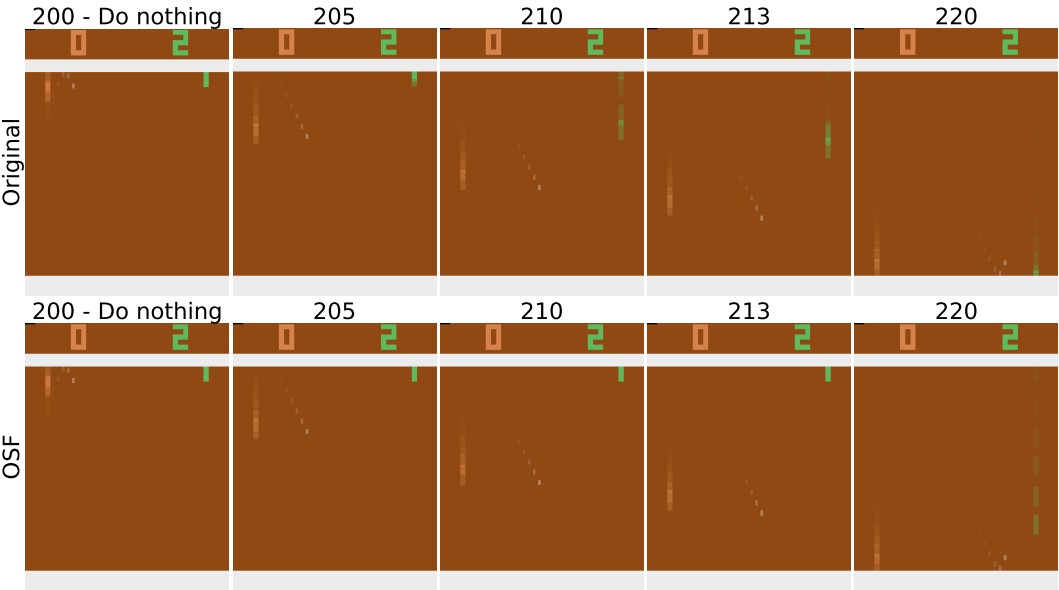

Figure 5: The numbers on top indicate the timesteps. Timestep 200 is the query state and timestep 213 is the OSF state. The sequence of images shows the movement of the policy from the query state and onward. For the OSF explanation, the policy (green paddle) keeps doing nothing until the state after the OSF state at the timestep 213. The threshold value is $\delta = 0.05$.

internally as they are the most similar parts of the counterfactual states. However, it is difficult to use counterfactuals because the states are similar, yet it is not clear what decides the action selection. Hence, the counterfactual explanation lack discriminative power.

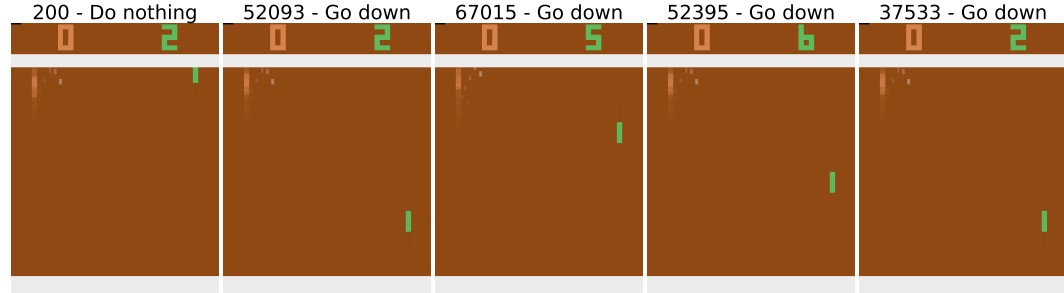

Figure 6: The numbers on top indicate the timesteps with the corresponding action for the state. The timestep is not reset when an episode ends. Timestep 200 is the query state. The rest of the states are counterfactual states retrieved based on closeness in the embedding space of the policy.

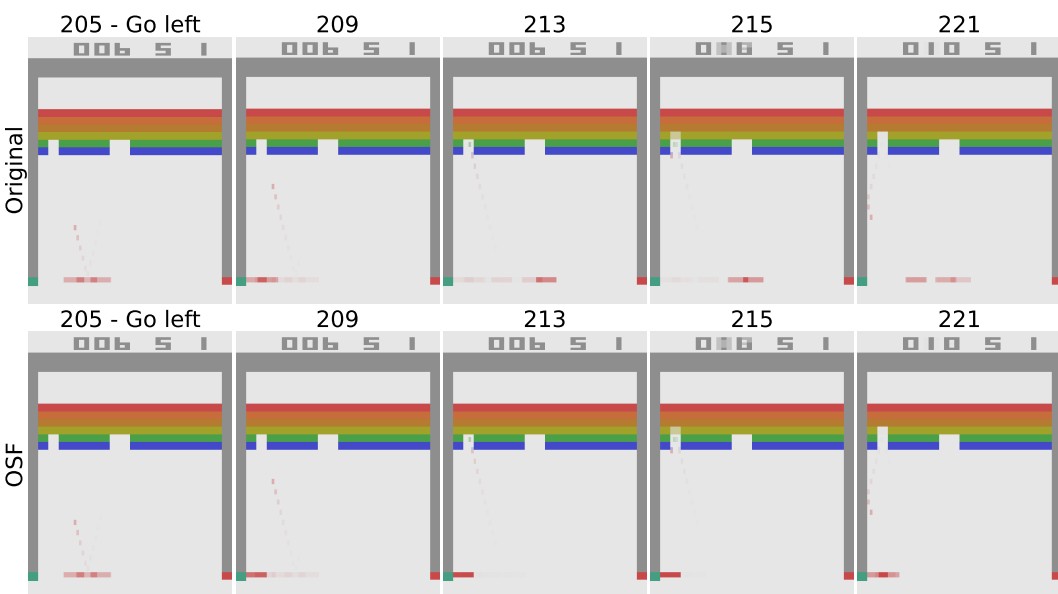

Figure 7: The numbers on top indicate the timesteps. Timestep 205 is the query state and timestep 215 is the OSF state. The sequence of images shows the movement of the policy from the query state and onward. For the OSF explanation, the policy (red paddle) keeps going left until the state after the OSF state at the timestep 215. The threshold value is $\delta = 0.05$.

### 5.2.3 BREAKOUT

We run the same setup for Breakout like Pong to showcase the value of OSF explanations. We changed the color of the game background to ease visualization post hoc but did not alter the input given to the agent. Figure 7 demonstrates that when the ball leaves the paddle, it is unnecessary to make any movement. However, without our method, we would not know that the additional movements made by the agent are unneeded. Also, they increase the complexity for a stakeholder trying to understand the agent.

Fig. 8 shows the same query state as in Fig. 7 but with counterfactual states similar to the query state. These counterfactual states are hard to use as they do not pinpoint specific game elements triggering an action. Moreover, the counterfactual explanations assume that by observing them, the stakeholder should understand the behavior which we do not believe works well in RL. Finally, the states are similar, yet a different action is triggered. Much of the issue is caused by the decision regions being small. The decision regions in RL are not like in supervised learning where the decision boundary can be far away from the query state.

Table 1: The performance of policies averaged over 30 episodes. The furthest left column refers to the environment in which the experiment took place. Original refers to the deep Q-network (DQN) policy without any modification. Simplified is the version where we set the starting state as the query state, find the OSF state, set the next state as the query state, and continue with the same procedure until termination. The same underlying policy is used in both. $\epsilon$ is the probability of executing a random action in the DQN, that is, the $\epsilon$ in the $\epsilon$-greedy algorithm. $\delta$ is the stopping criterion in Eq. (2). #Timestep/action switch refers to how many timesteps on average the policy executes the same action consecutively before it switches the action.

| | Method | $\epsilon$ | $\delta$ | Episodic Return ↑ | #Timestep/Action Switch ↑ |
|---|---|---|---|---|---|
| Pong | Original | 0.00 | | **21.00** | 7.70 ± 0.14 |
| | | 0.05 | | 18.67 ± 1.88 | 7.25 ± 0.19 |
| | Simplified | | 0.02 | **21.00** | 64.66 ± 7.16 |
| | | | 0.03 | **21.00** | 75.12 ± 1.84 |
| | | | 0.04 | **21.00** | 73.27 ± 3.74 |
| | | | 0.05 | **21.00** | **75.96 ± 6.21** |
| Pacman | Original | 0.00 | | 1423.33 ± 331.65 | 12.79 ± 0.95 |
| | | 0.05 | | 1917.67 ± 787.84 | 10.92 ± 1.09 |
| | Simplified | | 0.02 | 1966.00 ± 185.22 | 19.21 ± 1.37 |
| | | | 0.03 | 2119.00 ± 386.20 | 21.59 ± 2.25 |
| | | | 0.04 | 2037.00 ± 208.21 | 24.38 ± 1.82 |
| | | | 0.05 | **2363.33 ± 639.23** | **29.48 ± 3.72** |
| Space Invaders | Original | 0.00 | | 1231.00 ± 182.63 | 10.17 ± 0.18 |
| | | 0.05 | | 990.00 ± 623.91 | 9.06 ± 0.68 |
| | Simplified | | 0.02 | **1579.33 ± 1030.24** | 18.20 ± 1.49 |
| | | | 0.03 | 1293.50 ± 370.62 | 23.90 ± 2.03 |
| | | | 0.04 | 1467.83 ± 876.03 | 24.67 ± 2.24 |
| | | | 0.05 | 724.50 ± 197.60 | **27.29 ± 1.65** |
| Assault | Original | 0.00 | | 1681.23 ± 350.44 | 8.18 ± 0.27 |
| | | 0.05 | | 1315.00 ± 271.65 | 7.63 ± 0.21 |
| | Simplified | | 0.02 | 1833.23 ± 481.17 | 11.57 ± 0.41 |
| | | | 0.03 | 2652.50 ± 717.39 | 14.10 ± 1.42 |
| | | | 0.04 | 2997.13 ± 2109.81 | 15.69 ± 1.59 |
| | | | 0.05 | **3377.30 ± 1416.44** | **16.91 ± 1.19** |
| Seaquest | Original | 0.00 | | 2928.00 ± 459.85 | 8.13 ± 0.69 |
| | | 0.05 | | 1836.00 ± 568.81 | 7.93 ± 0.46 |
| | Simplified | | 0.02 | 2238.67 ± 316.79 | 13.41 ± 0.54 |
| | | | 0.03 | 2776.00 ± 759.59 | 14.33 ± 0.65 |
| | | | 0.04 | **4088.00 ± 1697.26** | 15.17 ± 1.16 |
| | | | 0.05 | 1103.33 ± 833.75 | **19.03 ± 2.69** |
| Breakout | Original | 0.00 | | 382.20 ± 3.50 | 13.77 ± 3.24 |
| | | 0.05 | | 313.97 ± 106.97 | 8.24 ± 0.51 |
| | Simplified | | 0.02 | **387.50 ± 24.05** | 19.61 ± 4.48 |
| | | | 0.03 | 308.47 ± 139.57 | 31.53 ± 18.02 |
| | | | 0.04 | 375.23 ± 29.42 | **35.73 ± 12.66** |
| | | | 0.05 | 314.37 ± 44.90 | 23.61 ± 4.27 |

## 5.3 POLICY PERFORMANCE WITH OSF

Until now, we have seen how OSF explanations work qualitatively. To show quantitative results, we roll out policies using our method to show that the method results in less action switching. Action switching refers to the act of executing a different action $a_t$ in the current state compared to the action $a_{t-1}$ in the previous state, that is $a_t \neq a_{t-1}$. We set the starting state as the query state and execute the query state action until a threshold is reached, following Eq. (2). When the threshold is reached, we set the state after the OSF state as the query state and re-

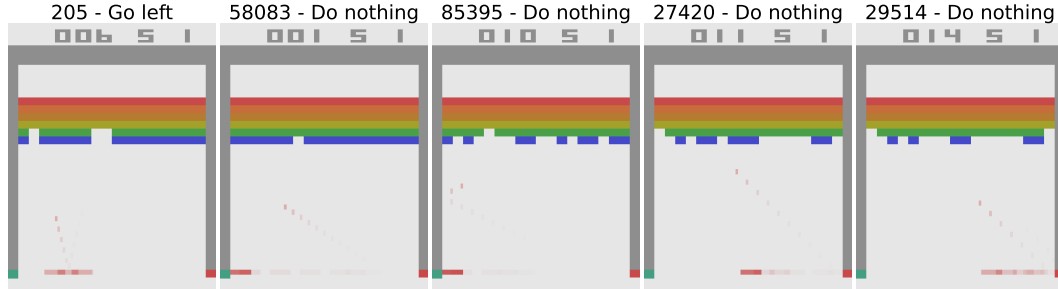

Figure 8: The numbers on top indicate the timesteps with the corresponding action for the state. The timestep is not reset when an episode ends. Timestep 205 is the query state. The rest of the states are counterfactual states retrieved based on closeness in the embedding space of the policy.

peat the process. This process repeats until an episode ends. To check whether less action switching happens, we divide an episode's length by the number of times the agent switches actions. We use the same set of $\delta$ values across all Atari environments because CleanRL uses `stable_baselines3.common.atari_wrappers.ClipRewardEnv` that bins the reward to the set $\{-1, 0, 1\}$ depending on the sign of the reward that the agent receives (Raffin et al., 2021; Huang et al., 2022).

Table 1 shows that we can reduce the action switching by at least a factor of two across six environments. Additionally, we observe that the performance of the policy in terms of return is not negatively affected and even improves in some environments. Surprisingly, the performance increases even though we naively set the query states. These results support the case studies that show there are situations where the agent moves unnecessarily. These extra movements add to the complexity of interpreting the agent without performance gain. Also, the results align with the thought that in RL, not all states have the same importance.

## 6 DISCUSSION AND CONCLUSION

We presented a new method to understand the long-term behavior of RL agents. The method finds outcome-based semifactual (OSF) explanations that focus on explaining the outcome rather than actions. Given a state of interest, called the query state, the OSF explanation asks how long an agent can execute the same action before it *has to* switch due to a negative outcome. We achieve this by simulating trajectories and keeping track of the expected return if the agent keeps using the policy versus executing the same action. When the difference between these two estimates reaches a threshold, we stop and set the state as the OSF state. Like state importance methods, we show the entire rollout from the query state to the OSF state to better understand the agent's behavior. To demonstrate the usefulness of our method, we qualitatively show examples of explanations produced by our method in three environments. The results show many situations where the agent unnecessarily changes actions, increasing the complexity of understanding its behavior. To emphasize that the usefulness extends beyond these few examples, we quantitatively show how our method can improve policy performance while reducing the number of action switches across six environments.

Our method depends on selecting effective query states. However, the same applies to most local explanations, such as counterfactuals or saliency maps. Another limitation of our method is the need for a state transition probability function. This can be improved by training world models. Because our method does not need to forecast far into the future, we believe small compounding errors in the state transition probability model should not significantly impact our method. Finally, our method can be computationally intensive. We need to have two environments initialized to the same query state for rollouts and comparisons.

In the future, we should perform evaluations through user studies and consider how they should be set up so that the results are comparable to other works. Finally, we should investigate how using world models affects the method.

REPRODUCIBILITY STATEMENT

We provide code and trained models at `https://anonymous.4open.science/r/osf-explanation-for-rl-E312/`. The repository includes a PIP requirement file that contains the name of packages used and their versions. The hyperparameters used for each experiment are detailed in the paper. Additionally, the code used to create the figures in the paper is included. The data used is generated from the code itself using Gymnasium (Towers et al., 2024) and does not need to be externally downloaded. All experiments ran on a MacBook Pro 2023, Apple M2 MAX, 64 GB RAM. We used Python v3.11.8 and PyTorch v2.2.2 (Ansel et al., 2024) with the MPS backend for GPU accelerated training.

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

# A APPENDIX

## A.1 ALGORITHM

Algorithm 1 shows the full algorithm of our method described in Section 4.

---

**Algorithm 1** Find OSF state $s'_{t+n}$

---

**Input**: $s_t$: query state, $\delta$: stopping criterion, $Q$: state-action value function,
    $T$: state transition probability function, and $IG$: importance gap estimation function.
**Output**: $s_x$: outcome-based semifactual state.

1: **procedure** FIND_OSF($s_t, \delta, Q, T, IG$)
2:     $z \leftarrow 0; i \leftarrow 0; s'_t \leftarrow s_t$
3:     **while** $z < \delta$ **do**
4:         $a_{t+i} \leftarrow \arg\max_a Q(s_{t+i}, a)$          ▷ Get action for trajectory $\tau$
5:         $s_{t+i+1} \sim T(\cdot \mid s_{t+i}, a_{t+i})$          ▷ Get next state for trajectory $\tau$
6:         $s'_{t+i+1} \sim T(\cdot \mid s'_{t+i}, a_t)$          ▷ Get next state for trajectory $\tau'$
7:         $z \leftarrow IG(s_{t+i+1}, s'_{t+i+1} \mid a_t)$          ▷ Compute importance gap
8:         $i \leftarrow i + 1$
9:     $s'_{t+n} \leftarrow s'_{t+i-1}$
10:     **return** $s'_{t+n}$

---

## A.2 ADDITIONAL QUALITATIVE RESULTS

In this section, we show additional results. Fig. 9 shows similar behavior as in Fig. 5, where many unnecessary actions are abstracted away so that we can focus on those that matter. Fig. 9 displays a query state where the policy can keep doing nothing for 50 timesteps without receiving less reward. Again, it allows us to see the boundary between different outcomes, one where we do not lose a point and the other where we do lose a point. It is possible to draw some insights from the counterfactuals in Fig. 6. For example, one might hypothesize that the policy's position determines the action since all go up and all go down examples show similar paddle positions. Fig. 11 shows an additional OSF explanation for Breakout. Fig. 12 illustrates the corresponding counterfactual explanations for Breakout.

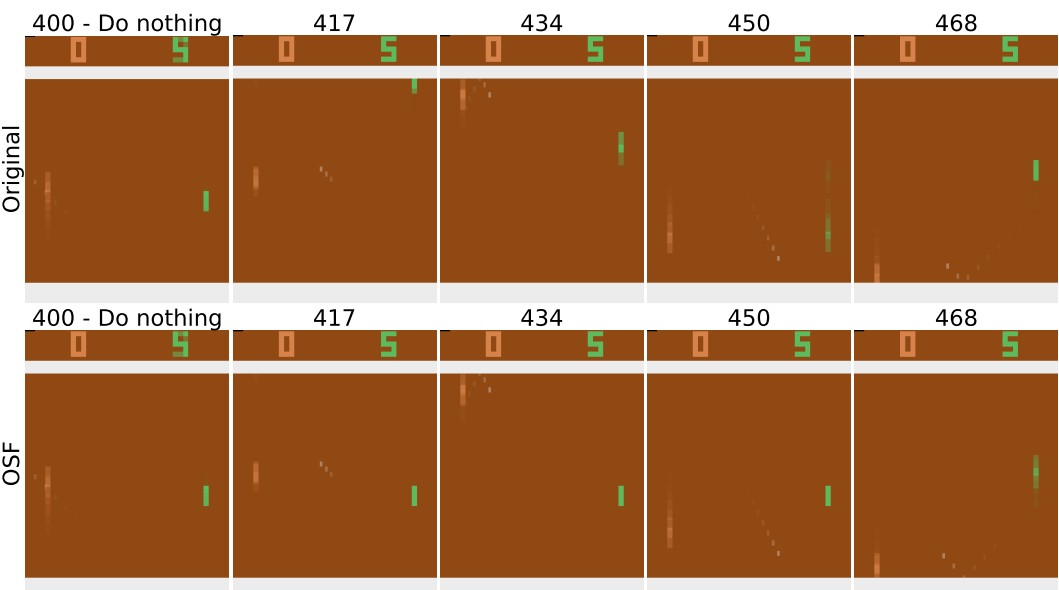

Figure 9: The numbers on top indicate the timesteps. Timestep 400 is the query state and timestep 450 is the OSF state. The sequence of images shows the movement of the policy from the query state and onward. For the OSF explanation, the policy (green paddle) keeps doing nothing until the state after the OSF state at the timestep 450. The threshold value is $\delta = 0.05$.

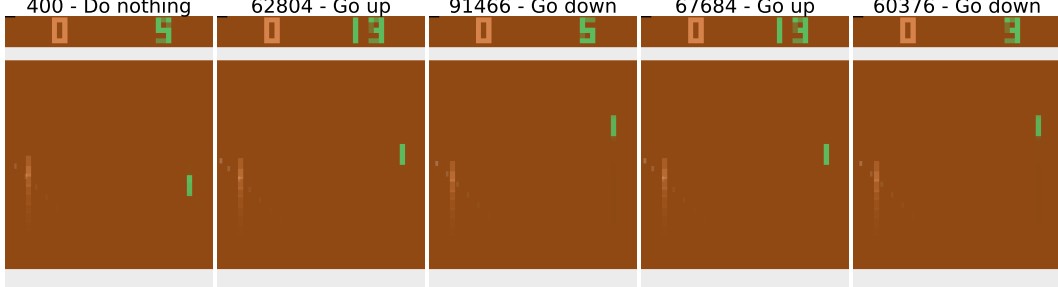

Figure 10: The numbers on top indicate the timesteps with the corresponding action for the state. The timestep is not reset when an episode ends. Timestep 400 is the query state. The rest of the states are counterfactual states retrieved based on closeness in the embedding space of the policy.

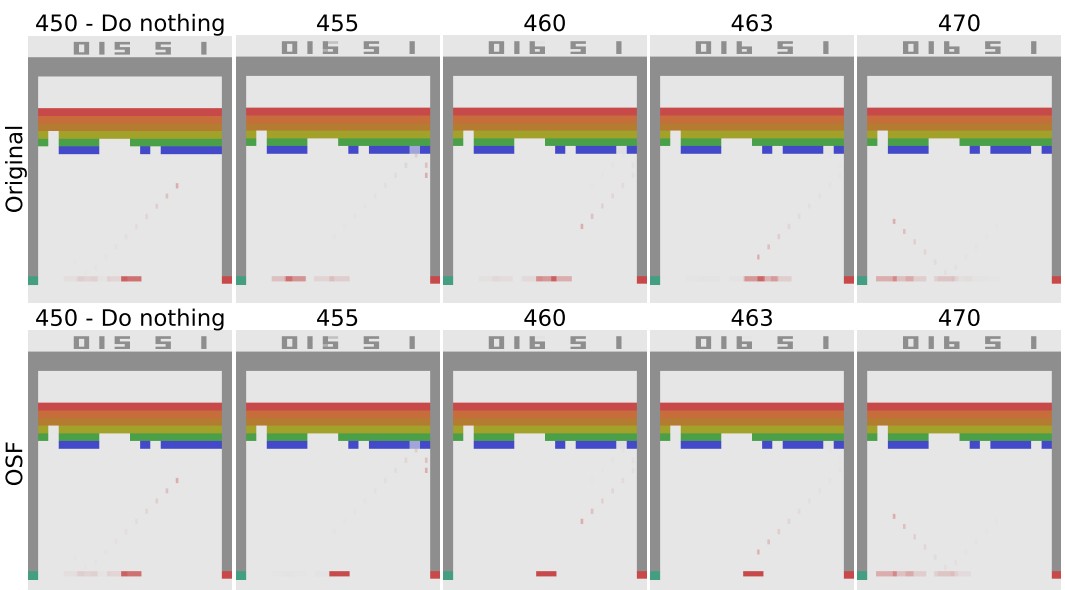

Figure 11: The numbers on top indicate the timesteps. Timestep 450 is the query state and timestep 463 is the OSF state. The sequence of images shows the movement of the policy from the query state and onward. For the OSF explanation, the policy (red paddle) keeps doing nothing until the state after the OSF state at the timestep 463. The threshold value is $\delta = 0.05$.

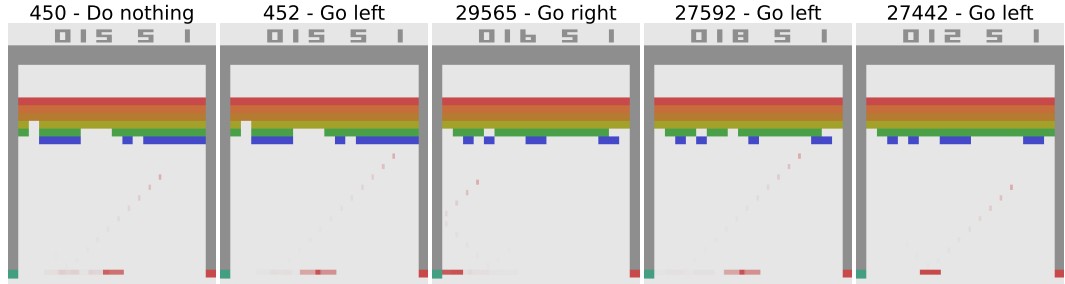

Figure 12: The numbers on top indicate the timesteps with the corresponding action for the state. The timestep is not reset when an episode ends. Timestep 450 is the query state. The rest of the states are counterfactual states retrieved based on closeness in the embedding space of the policy.

