# OpenReview forum: "Outcome-based Semifactual Explanation For Reinforcement Learning"
_ICLR.cc/2025/Conference — ICLR 2025 Conference Withdrawn Submission_

### Official Review · Reviewer_3wSM · 2024-10-16

**Soundness:** 2
**Presentation:** 2
**Contribution:** 2
**Rating:** 1
**Confidence:** 4

**Summary:**

This paper introduces OSF explanations for RL, designed to address the challenges of defining similarity, handling out-of-distribution states, and improving discriminative power in counterfactual explanations. The method identifies the last state where an agent can continue taking the same action as the query state without negative outcomes, offering a more intuitive understanding of the agent's behavior. The authors claim that their approach enhances policy performance and reduces action-switching across multiple environments, providing both qualitative and quantitative benefits.

**Strengths:**

There are no significant strengths to highlight in this paper. While the concept of OSF explanations is intriguing, it is not clearly enough explained or justified.

**Weaknesses:**

The paper suffers from several critical weaknesses:

The proposed method is not sufficiently clear. The explanation of how OSF is derived and how it solves the outlined challenges lacks clarity, making it difficult to follow the technical details and understand the novelty of the approach.

The results are not presented in a manner that clearly demonstrates the advantages of OSF explanations. It is unclear how the method performs in comparison to existing techniques or whether the claimed improvements are meaningful.

The overall contribution of the paper is not well articulated. The reader is left questioning how the proposed approach significantly advances the state of the art in RL explanations.

The writing requires significant improvement, as it currently makes the paper hard to follow and detracts from the comprehension of the method and results.

**Questions:**

N/A

---

### Official Review · Reviewer_JDYQ · 2024-11-01

**Soundness:** 2
**Presentation:** 2
**Contribution:** 2
**Rating:** 3
**Confidence:** 4

**Summary:**

The author stated that they developed a novel method for analyzing long-term behavior in RL agents through outcome-based semifactual (OSF) explanations. This method concentrates on outcomes instead of actions and involves simulating trajectories from a designated query state to determine how long an agent can sustain an action before being compelled to change due to adverse outcomes. Additionally, the author noted that this approach reveals instances where agents unnecessarily alter their actions, thus complicating the understanding of their behavior. They also implemented this method in three different environments.

**Strengths:**

The author introduced a new algorithm designed to offer insights into outcomes. This approach is straightforward and easy to implement.

**Weaknesses:**

1. The motivation behind the presented explanation method is not clear to me. I am uncertain about the necessity of such explanations, and the author does not adequately address how this approach contributes to explainability or how it could help users or stakeholders better understand the decision-making processes of RL agents.

2. The paper lacks theoretical analysis and does not provide a detailed discussion about the outcomes. For instance, the author fails to justify why the simplified design is an improvement over the original DQN.

3. The focus of this paper appears scattered—it begins with an emphasis on explainable AI but shifts towards improved performance and action switching. This shift created a sense of inconsistency while reading. I believe the author needs to reconsider the structure of the paper to achieve greater coherence.

4. Some benchmark design, for example comparing the original model with e = 0.05,  seems irrelevant. They author need to clarify why they pick those benchmarks. At least, for testing and evaluation, the exploration effect should always be removed.

5. Rather than focusing solely on the explainability aspect of this work, I believe the methodology could be more valuable in scenarios where the cost of changing actions is high. This could be more applicable in real-world settings where such costs are significant.

**Questions:**

1. Why is OSF important? Understanding how long an agent can sustain an action before a negative outcome necessitates a change is interesting, but what are the broader implications? How does this knowledge deepen our understanding of RL? You mention that this aids in explainable AI, but it's essential to elaborate on why this information is vital.

2.  Why does the performance exceed that of the original DQN? Is the baseline model of the simplified version the same as the original? When comparing how much the simplified model deviates from the original, you need to justify why your proposed method is superior and provide relevant explanations or theoretical analysis.

3. What is the standard for you to pick the benchmarks you compare with? why you have e = 0.05 as a benchmark?

---

### Official Review · Reviewer_h8bd · 2024-11-03

**Soundness:** 2
**Presentation:** 3
**Contribution:** 2
**Rating:** 5
**Confidence:** 3

**Summary:**

The authors propose a novel method called Outcome based Semi-Factual (OSF) explanation which allows them to explain an RL agent’s decisions as well as improve policy performance. Their basic idea is as follows. Given an input state called query state, they identify the optimal action to be taken at this state and also for how many more states in this trajectory sequence can the same action be taken without incurring a significant loss in performance. The final state in this sub-trajectory is called the OSF state for this query state. The authors claim that this approach yields better explanations for RL compared to counterfactual explanation methods  and also show that this leads to better performance with lesser switching between actions in six Atari environments and in the mountain car environment. The contributions in this paper are 1) OSF explanations 2) study of effectiveness of OSF explanations using numerical examples and 3) using OSF to minimize action switching along trajectories.

**Strengths:**

1. Clearly written paper – problem and solution approach are explained well.
2. The problem considered is an important one in RL.
3. To the best of my knowledge, definition and use of OSF states for explanation and policy improvement are novel and interesting.
4. The approach presented by the authors ensures that any state used for explanation is in the reachable set of states given a start state and policy; unlike generative methods for counterfactual states, which may lead to generating out-of-distribution or sometimes even invalid states.

**Weaknesses:**

1. The authors state in Line 230 that: “To find the OSF state, we assume access to a Q-function and a state transition probability function T that enables us to simulate trajectories.” Doesn’t this require addition learning machinery and more importantly additional data which can be expensive? Specifically, if T is not given in the problem, won't it have to be learned using data sampled from an exploration policy?
2. The authors restrict attention to deterministic state transition functions in their experiments. In many practical problems in RL the state transition functions are stochastic and hence this would be a crucial limitation. Can the authors discuss as to what modifications (if at all) would be needed in extending their approach to stochastic environments?
3. The authors assume the same feasible set of actions for all states. While this is a standard assumption in several RL problems, I think a discussion on this limitation will be useful, especially for real-world problems.
4. Finding a good $\delta$ value to determine OSF states is dependent on the environment and also requires exploration by a knowledgeable human stakeholder. This can be an additional cost in real-world use cases. Can the authors add a discussion or a suggestion on ways to find a good $\delta$?
5. While the approach is interesting, based on the experiments presented in this paper, it is not clear if this approach will scale to larger practical problems (i.e. problems with much larger state and action spaces). Can the authors discuss the computational complexity of their approach, especially with respect to scaling to much larger environments. An example of a larger environment would be a self-driving car simulator or Minecraft etc.

**Questions:**

1. Since this is a post-hoc explanation method, how does this impact policy performance? Is the policy updated based on the OSF explanation?
2. What is the sensitivity to the accuracy of learned Q-values?
3. When the authors define an OSF explanation tuple as $\langle s_t, s^\prime_{t+n} \rangle$, and the condition $max_{a}(Q(s_{t+n}, a)-Q(s_{t+n}, a_t)) \le \delta$, while the authors condition that this value gap of $\delta$ will not be breached in any of the states between $t$ and $t+n$ (from equation (2)). However, shouldn’t the authors consider some cumulative performance loss here?
4. How can the authors justify this statement in line 225: “Additionally, the OSF state is visually similar to the query state given that $\delta$ is not too large.”
5. Is this statement mentioned in Line 508: “Additionally, we observe that the performance of the policy in terms of return is not negatively affected and even improves in some environments” due to the fact that the original RL policy was sub-optimal and not due to any stochasticity? Can the authors provide some additional analyses or ablation to clarify this?
6. Can ideas from the literature of semi-MDPs and options be combined with the authors’ approach?
7. Will adding a switching cost or penalty to the reward function yield similar policies?

**Details Of Ethics Concerns:**

The ethical concerns are only limited to the concerns for any RL algorithm: that caution must be exercised regarding potential biases and harmful outcomes when using these in real-world use cases.

---

### Official Review · Reviewer_Mb42 · 2024-11-04

**Soundness:** 3
**Presentation:** 3
**Contribution:** 2
**Rating:** 5
**Confidence:** 2

**Summary:**

This paper introduces Outcome-based Semifactual (OSF) explanations to explain the behavior of RL agents. The authors argue that existing methods for explaining RL agents, such as counterfactual explanations and state importance methods, have limitations, particularly in their ability to capture the nuances of sequential decision-making and long-term outcomes. OSF addresses these limitations by focusing on how long an agent can maintain its current action in a given state without incurring a significant negative impact on the expected return. The method works by simulating trajectories from a query state, both using the agent's policy and by forcing the agent to repeat the action taken in the query state. The OSF state is defined as the last state in the forced-action trajectory where the expected return, as measured by the importance gap, remains similar to the policy-guided trajectory. This state is then used as an explanation (of type "even-if").

By analyzing OSF states on Mountain Car and Atari game domains, the authors demonstrate that RL agents often switch actions unnecessarily, leading to more complex behavior without improving performance. They claim that counterfactual states generated using methods like nearest neighbor search, while similar to the query states and sometimes helpful in identifying features the policy focuses on, often lack the discriminative power to fully explain the policy's behavior. Furthermore, they show that by starting from an initial state, executing the action chosen by the policy until the next OSF state and repeating the same process from the state that follows, it is possible to significantly reduce the action switches without affecting the return negatively.

**Strengths:**

Semifactual explanations are not a new idea. They have been explored in various fields, including philosophy, psychology, and more recently, AI research on explainable machine learning. In settings where decision boundaries are more straightforward to define, such as classification tasks, semifactual states typically do not cross the decision boundary. This characteristic highlights that traditional semifactual explanations often focus on exploring state variations that maintain the same immediate outcome, for example, staying within the same class label.

In contrast, Outcome-based Semifactual (OSF) states, as defined by the authors can extend beyond the decision boundary for actions, as long as they maintain a similar (future) outcome. The explicit distinction between the immediate decision outcome (the action taken at a state, which remains the same for the query and the semifactual states) and the future outcome (i.e. the expected return) is novel. The focus is more on the boundary of acceptable deviations from the policy's expected return, as denoted by the importance gap threshold. Taking the same action as long as the gap is below the threshold is a simple way to find an OSF explanation that leads to a trajectory that is easy to interpret in terms of the agent's behavior.

**Weaknesses:**

The prior work on semifactual explanations in RL (Gajcin et al., 2024) defines desired properties for this type of explanation and proposes algorithms for finding semifactual states that satisfy these properties as much as possible. In the paper, although the authors state their intent (i.e., understanding how much a query state can be modified by executing the same action without affecting the future outcome negatively), it is not fully articulated why identifying such a state would provide a valuable explanation for the agent's behavior. Examining the role of stochastic uncertainty, fidelity (especially its relationship to the importance gap threshold), and exceptionality in shaping the informativeness of the resulting OSF state could address this gap in the paper's reasoning.

Despite the limitations of using nearest neighbors for generating counterfactual explanations, as mentioned in the paper, this approach does offer some robustness against stochasticity in the environment. The proposed method for finding the OSF states, which relies on executing the same action may be more susceptible to a higher importance gap in the transitioned states and consequently more frequent action switches under such a setting. While the paper acknowledges the potential impact of stochasticity on OSF states, the experimental evaluation focuses on environments with deterministic state transitions. Addressing the applicability of the method in stochastic environments would strengthen the paper's conclusions.

**Questions:**

- Under the discounted reward setting (as mentioned in section 2), the importance gap will usually be smaller at the beginning of an episode and will get higher at the end (e.g. when the reward is sparse or defined to encourage the agent to reach the goal as quick as possible, as in the Mountain Car domain). The stopping criterion \delta on the other hand is fixed and global. Can you elaborate on the consequences of this? (e.g. on the # timestep / action metric)

- In figure 4, both the query states and the \delta values change. How sensitive are the results and interpretations? Plotting the OSF states for the same query state and different \delta values can provide valuable insights into how the OSF explanation changes as the allowed deviation from the policy increases. Also, in table 1, we can observe that although Pong is not sensitive to \delta, some other games are; similar qualitative results for them would also be useful and can help to clarify the changes in the episode returns.

---

### Note · Authors · 2024-11-15

I have read and agree with the venue's withdrawal policy on behalf of myself and my co-authors.